# An In Vivo Rat Study of Bioresorbable Mg-2Zn-2Ga Alloy Implants

**DOI:** 10.3390/bioengineering10020273

**Published:** 2023-02-20

**Authors:** Alexey Drobyshev, Zaira Gurganchova, Nikolay Redko, Alexander Komissarov, Viacheslav Bazhenov, Eugene S. Statnik, Iuliia A. Sadykova, Eugeny Sviridov, Alexey I. Salimon, Alexander M. Korsunsky, Oleg Zayratyants, Denis Ushmarov, Oleg Yanushevich

**Affiliations:** 1Laboratory of Medical Bioresorption and Bioresistance, Moscow State University of Medicine and Dentistry, 127473 Moscow, Russia; 2Laboratory of Hybrid Nanostructured Materials, National University of Science and Technology “MISiS”, 119049 Moscow, Russia; 3Casting Department, National University of Science and Technology “MISiS”, 119049 Moscow, Russia; 4HSM Laboratory, Center for Digital Engineering, Skoltech, 121205 Moscow, Russia; 5Trinity College, Oxford OX1 3BH, UK; 6Laboratory of the Clinical Medical Center, Moscow State University of Medicine and Dentistry, 111398 Moscow, Russia; 7Educational and Production Department, Kuban State Medical University, 350912 Krasnodar, Russia

**Keywords:** bioresorption, magnesium alloys, bioresorbable materials

## Abstract

In the present study, pins made from the novel Mg-2Zn-2Ga alloy were installed within the femoral bones of six Wistar rats. The level of bioresorption was assessed after 1, 3, and 6 months by radiography, histology, SEM, and EDX. Significant bioresorption was evident after 3 months, and complete dissolution of the pins occurred at 6 months after the installation. No pronounced gas cavities could be found at the pin installation sites throughout the postoperative period. The animals’ blood parameters showed no signs of inflammation or toxication. These findings are sufficiently encouraging to motivate further research to broaden the experimental coverage to increase the number of observed animals and to conduct tests involving other, larger animals.

## 1. Introduction

The history of employing metallic structures for bone repair has witnessed more than 150 years of selection and application of various metal alloys for the fabrication and optimization of bone fixators [1,2,3]. The first attempts to use magnesium alloy implants for bone fixation made in the beginning of the 21st century were unsuccessful [4,5]. Magnesium-based alloys developed in recent years have demonstrated improved corrosion resistance and mechanical properties and are promising materials for creating biodegradable, biocompatible metal implants [6]. In clinical practice to date, components made from titanium alloys are the most often used implants [7]. The present-day market for medical devices for osteosynthesis is occupied by implants made from non-resorbable (bioinert) metals and alloys. While the non-degradable implants for replacing bone defects are made of stainless steel or titanium and provide the maximum level of stability, they also have significant disadvantages, such as X-ray screening, the possible development of inflammatory changes around them, etc. In contrast, biodegradable (bioresorbable) magnesium alloys show promising properties due to the inherent ability of magnesium and its alloys to decompose without releasing toxic corrosion products. This has led to a wide range of applications in the biomedical field, including cardiovascular stents and fixation structures for osteosynthesis [3,4]. The main obstacle to the wider use of this material is linked to its somewhat limited mechanical properties, which may lead to serious problems in bone remodeling [8,9], and the possibility of the release of toxic ions and microparticles as a result of corrosion and material disintegration, which may cause inflammatory osteolysis [10,11]. With the prolonged use of metal fixing structures in the epiphyses of bones, a high concentration of metal particles is found in the synovial fluid and tissue around the structure as a result of the continuous release of metal particles under mechanical stress [12]. In most cases, the use of fixing structures made of bioinert metals during osteosynthesis and the absence of their biodegradation requires repeated surgical intervention aimed at removing metal structures that have fulfilled their role, and, often, this is a no less traumatic process than osteosynthesis itself, which entails an increase in the total duration of the related hospital stay and treatment and may cause temporary disability in patients. It is worth noting the limited use of such structures in children and adolescents due to their body growth, as well as the possibility of bacterial contamination of such bioinert metal implants. Although non-degradable metal implants are generally considered non-toxic, some of their components may contribute to the development of neoplasia [13]. Cases of osteogenic sarcoma development were noted in patients after the implantation of metal endoprostheses [14]. Additional disadvantages include the impossibility of using titanium implants in cancer patients, as this may be associated with the development of complications during radiation and chemotherapy. Thus, there is a need to develop materials for new-generation implants that possess the necessary strength and that undergo bioresorption, obviating the need for surgical intervention to remove them [15].

To date, there are three main groups of biodegradable materials that can be used for osteosynthesis: polymers, ceramics and their composites, and the bioresorbable alloys of various metals [5]. Magnesium-based alloys belong to the last group of materials and offer a number of advantages over bioinert metal alloys, polymers, and bioceramics.

Magnesium is a vital chemical macronutrient (comprising 0.2% of a human’s body weight) deposited in the bone tissue that is considered to be non-toxic and has good biocompatibility, biodegradability, and absorbability, as well as high tensile strength, compared with polymers, and it is less stiff than ceramics. The elastic modulus of magnesium (45 GPa) is closer to the elastic modulus of cortical bone (15–25 GPa) in comparison with the elastic moduli of titanium alloys and stainless steel (115–200 GPa) [16,17].

When biodegradable implants are installed, their degradation rate must not exceed the tissue growth rate to ensure proper filling of the voids formed within the implant by the new bone tissue. The addition of alloying elements into Mg alloys or the use of barrier coatings to reinforce bone growth can improve the bone healing process [18]. For example, the addition of Ga to the hydroxyapatite coating on Gription™ implants (West Chester, PA, USA) leads to a two-fold increase in the bone growth rate in beagle dog bones [19]. This was achieved by the inhibition of bone resorption by the Ga ions [20,21], which has been shown to be effective in the treatment of osteoporosis [22], hypercalcemia [23,24,25], Paget′s disease [26,27], and multiple myeloma [28].

The studied Mg-2Zn-2Ga (wt.%) alloy has previously shown a low in vitro corrosion rate and low cytotoxicity [29], and it was chosen for a further investigation of its in vivo corrosion rate on an animal model (Wistar rat). Many studies have shown a positive effect of magnesium biodegradation products on osteogenesis, but the mechanism of their action remains unclear [30,31,32,33,34]. According to one theory, certain proteins are adsorbed on the surface of such material from the biological environment, stimulating the growth of bone cells and the healing process. This is preceded by ion exchange reactions on the interstitial surface and the appearance of a magnesium phosphate layer [35]. It is believed that this reaction promotes the formation of direct chemical bonds between the magnesium implant and the mineral phase of the newly formed bone tissue. Ideal structures for bone fixation should have a lower rate of resorption compared with the process of bone remodeling. Biodegradable magnesium alloys can achieve a synchronization of the changes in their strength and the restoration of bone tissue. In contrast, the mechanical properties of permanent implants made of titanium and stainless steel remain virtually unchanged during the entire process of bone defect healing, and they may cause uneven bone remodeling that manifests itself as a combination of resorption sites with bone tissue hypertrophy [36,37].

However, despite the significant advantages, a number of unresolved issues persist that relate to the prospective use of bioresorbable magnesium alloys. Pure magnesium and some of its alloys corrode too rapidly under physiological conditions, leading to the early loosening or disintegration of implants before new bone has formed. Rapid corrosion causes excessive hydrogen release at the implant site, which can have a negative effect on the surrounding tissue and prevent bone regeneration [38,39,40]. The addition of various alloying elements can affect the amount of hydrogen released. Interestingly, some studies have reported that the addition of Ca as an alloying element increases the rate of degradation, increases the pH level, and increases the amount of hydrogen released during degradation [30]. It has also been described that the release of hydrogen may have a beneficial effect since hydrogen has antioxidant activities and can act as an absorber of hydroxyl radicals and peroxynitrite [8].

In summary, despite the great potential of magnesium and its alloys as materials for biodegradable implants, the main obstacle to their wider use consists of their rapid and uncontrolled degradation in the physiological environment, which is accompanied by the release of hydrogen [41]. In some cases, these limitations can be overcome by the careful selection of the chemical composition of the alloy and its thermomechanical treatment, e.g., by adding Ga (gallium) as an alloying element to the Mg alloy. Gallium ions have been clinically proven to be effective against bone resorption and are used to treat osteoporosis and hypercalcemia [18,19].

The purpose of the present study was to assess the effectiveness of a new Mg alloy bioresorbable implantation system via an experimental study of laboratory animals to assess the biocompatibility, rate of resorption, and influence of the Mg alloy’s degradation products on the animal′s health.

## 2. Materials and Methods

### 2.1. Mg-2 wt.% Zn-2 wt.% Ga Alloy Sample Preparation

The Mg-2 wt.% Zn-2 wt.% Ga alloy samples were prepared using high-purity metals. Melting was carried out in a resistance furnace in a steel crucible coated with BN in a protective atmosphere of Ar plus 2 vol.% SF_6_. The resulting melt was purged with Ar before being poured into a preheated aluminum mold. The ingot was solution heat-treated at 300 °C for 15 h and then at 400 °C for 30 h and machined. Then, the billet was subjected to hot extrusion from 50 to 20 mm (at an extrusion ratio of 6). The pins with a diameter of 1.5 mm and a height of 5 mm were cut from the extruded rod and ground with emery bars. The full description of the alloy sample preparation can be found elsewhere [42].

### 2.2. Animal Experiment

In the training vivarium of the Kuban State Medical University (Krasnodar, Russia) in July 2021, an experimental study was conducted on laboratory animals (rats), which consisted of the installation of magnesium alloy implants (Mn-2 wt.% Zn-2 wt.% Ga) in the animal’s femoral bone, with further observation in the early post-operative period, X-ray examination, and histological examination. The experiment used randomly selected white laboratory rats of the Wistar line, of both sexes, aged from 6 months, with an average body weight of 340–400 g, as shown in Figure 1.

The study protocol was approved by the interuniversity ethics committee (number 04 of 15 April 2021), and it complied with the principles of the “European Convention for the Protection of Vertebrate Animals Used for Experiments or for Other Scientific Purposes” dated 18 March 1986. The operation was performed under general anesthesia with an intramuscular injection of Flexoprofen (VIC, Vitebsk, Belarus) 2.5% 10 mg per kg and Zoletil 100 (Virbac, Carros, France) 20 mg per kg. Brilocaine (Ferain, Moscow, Russia) 1:200,000 was used for the local anesthesia. Under the conditions of the experimental operating room and in compliance with the rules of asepsis and antisepsis, a skin incision was made in the area of the femur from the outside and the bone was isolated. Each animal underwent the installation of 3 implants in the body of the femur, each with a diameter of 1.7 mm and a length of 5 mm. The implants are shown in Figure 2. The operating area with the bone and implants is demonstrated in Figure 3.

The wound was sutured with Vicryl 4-0 (Ethicon, Raritan, NJ, USA). The postoperative area was treated with the antibacterial aerosol Terramycin (Zoetis, Parsippany, NJ, USA). Postoperative antibiotic therapy was performed using intramuscular injections of Convenia (Zoetis, Parsippany, NJ, USA). The duration of the experiment was 6 months. A total of 6 animals were involved in the experiment. Each day, the animals were examined for their general condition and the condition of the postoperative area. Withdrawal from the experiment was carried out 1 month, 3 months, and 6 months after the operation. At the end of each study period, the animals were withdrawn from the experiment by an intramuscular injection and overdose of Telazol (Zoetis, Parsippany, NJ, USA). Then, the femur was harvested for further X-ray and histological examination. The samples selected for histological examination were placed in a 10% neutral formalin. Before withdrawal from the experiment, blood and urine samples were taken to determine the concentration of magnesium ions in the blood serum, as well as the markers of inflammation. One animal was withdrawn from the experiment 1 month after a consolidated pathological fracture of the femur, with the fragments freely lying in the soft tissues being visualized. The installed pins were integrated into the bone tissue. Perhaps, during the osteotomy to form the implant bed, the strength of the femur had decreased due to its small size (Figure 4a,b).

### 2.3. Ultrasound Examination

Before removing an animal from the experiment, an ultrasound examination was performed in the area of the femur. An ultrasound study was performed on days 7 and 14 and at 1 month, 3 months, and 6 months after implantation using a Philips Affiniti 70 device (Philips Healthcare, Bothell, WA, USA) with a surface sensor (L 12-3).

### 2.4. X-ray Examination

At the Department of Radiation Diagnostics, MGMSU, named after A.I. Evdokimov, targeted radiography and cone beam computed tomography (CBCT) of the biopsy specimens were performed. Targeted X-ray examination was carried out using the intraoral X-ray apparatus “FOCUS” Kavo (Biberach an der Riss, Germany). After the animals were withdrawn from the experiment, the cone beam computed tomography of the femur was performed using the X-ray computed tomograph ORTHOPANTOMOGRAPH OP 3D Pro, KaVo (Biberach an der Riss, Germany) in the “endodontics” mode, i.e., at the maximum resolution (90 kV, 4.0 mA, 6.1 s, 225 mGy cm^2^).

### 2.5. Microfocus Computed Tomography (micro-CT) Examination

To study the biodegradation of the implants in vivo, microfocus computed tomography using a microfocus X-ray tomograph (ELTECH-Med, Saint-Petersburg, Russia) was performed. The micro-CT was carried out on the basis of the Saint-Petersburg Electrotechnical University ETU LETI (Saint-Petersburg, Russia). The voxel size was 50 µm. The resulting images were processed using CTVox software (Bruker BioSpin, Reinstetten, Germany).

### 2.6. SEM and EDX

The elemental analysis was performed according to the same protocol that was used in our previous study [43]. Firstly, samples were dried for 1 day at −84 °C and 0.01 mbar pressure using a FreeZone Labconco freeze dryer machine (CEST, Skoltech, Moscow, Russia). Next, they were carefully cleaned of residual dried tissues. In order to eliminate the charging effect during the image acquisition by a Tescan Vega3 SEM (HSM laboratory, Skoltech, Moscow, Russia), a thin conductive gold layer was sputtered using a Quorum Q150T Plus coater (AICF, Skoltech, Moscow, Russia). The images were obtained under the following conditions: a high voltage of 20 kV, a current of 100 pA, an SE regime, and an exposure time of 2 min under the frames accumulation mode. The chemical composition was measured by an Oxford Instruments EDX detector (Department of Physical Chemistry, NUST MISiS, Moscow, Russia) that was calibrated in alignment with the standard reference material (SRM) 2910b, obtained from the National Institute of Standards and Technology. The composition was extracted only from flat locations.

### 2.7. Histological Evaluation of Bone Fragments

In the laboratory of the Clinical Medical Center MSMSU, we conducted the histological analysis of the bone samples from each animal. The bone fragments were fixed in a buffered solution of 10% formalin (Biovitrum, Saint-Petersburg, Russia) for at least 24 h. No pathological changes were visualized macroscopically. Decalcification was performed in all cases. The procedure was performed in a standard Biodek-R solution (BioOptica, Milano, Italy) for 1–3 days at room temperature. After neutralization of the decalcified residues with ammonia water, the specimens were forwarded for staged dehydration according to the following standard protocol: 95% ethanol, xylene, and paraffin. Paraffin sections of 3–4 μm thick were made and processed according to standard procedures and stained with hematoxylin and eosin. In addition to studying sections in a light microscope, all micropreparations were scanned using a Leica Aperio 1000 instrument (Wetzlar, Germany).

## 3. Results

The animals were monitored daily in the early post-operative period to study their general condition. In the first days after the operation, slight edema and hyperemia were visualized in the postoperative area (Figure 5a,b).

All clinical signs resolved within 2 weeks after surgery. Other clinical signs of local inflammatory reactions did not appear during the primary period after implantation. In addition, there were no significant gas cavities that could be detected by observation and palpation. According to the results of blood and urine tests, no deviations from the norm were revealed.

### 3.1. Results of Laboratory Studies

The laboratory analyses of the rats 1 month after the installation of the bioresorbable pins showed no significant deviations from the reference values. We assume that those deviations of the indicators that were determined were associated with stress factors when taking blood and urine from the animals. It should be noted that according to the presented analyses, no inflammatory, allergic, or toxic components were determined.

### 3.2. Ultrasound

Before an animal was removed from the experiment, an ultrasound study was performed in the femur area to assess the formation of gas cavities from the release of H_2_ during the magnesium biodegradation. A gaseous layer was visualized in the soft tissues over the area of the installed implants after 3 and 6 months (Figure 6a,b).

### 3.3. X-ray Examination

According to the targeted radiography of the fragments of the femur of one rat (bred after 1 month), the non-degraded implants were visualized (Figure 7a). According to the targeted radiography of the femur after 3 and 6 months, the implants were not visualized (Figure 7b,c). The implants proved to be well-fixed, demonstrating successful primary osseointegration. Of course, microscopic penetration of the fibrous tissue (between the experimental screws and the adjacent bone) cannot be ruled out. No osteolytic changes were found around the experimental screws, indicating an inflammatory reaction.

### 3.4. Cone Beam Computed Tomography

According to the CBCT data, implants with partial signs of bioresorption in the fragments of the femur were visualized after 1 month (Figure 8a,b).

After 3 and 6 months, the implants were not visualized. The formed phenomena were well-visualized both 3 (Figure 9a,b) and 6 months (Figure 10a,b) after implantation.

All implants examined appeared to be in contact with the surrounding bone without any evidence of foreign body reaction or fibrointegration. In addition, there were no osteolytic changes and no signs of bone irritation adjacent to the experimental implants.

### 3.5. Micro-CT

Micro-CT visualized increased density on the non-resorbed implant after 1 month (Figure 11a,b) in comparison with a slight decrease in the average density of implants after 3 months (Figure 12a,b)According to the micro-CT images 6 months after the operation, the holes formed for the implants were visualized, though the implants themselves were not visualized (Figure 13a,b). The resorption rates of the implants are shown in Table 1.

### 3.6. Scanning Electron Microscopy Characterization

In order to provide significant comparisons between each case, bones were extracted from similar locations in the rats’ bodies. Figure 14 demonstrates their microstructures near the Mg-based implants after different periods of time, such as 1 month, 3 months, and 6 months. One month after implantation, the Mg-based shafts were detected inside the bones. However, they fell out after the freeze-drying stage, which showed low levels of osseointegration with the bones. After 3 and 6 months, the implants were not observed, which can be related to the physical bioresorption occurrence that was confirmed by EDX analysis.

A comprehensive article on the bone–remodeling interaction with Mg-based implants by SEM and EDX was successfully performed in our previous study [42]. The developed procedure was applied to the samples in the current study. The chemical compositions obtained from the 100 × 100 μm^2^ flat areas are summarized in Table 2. EDX analysis revealed the presence of magnesium elements inside the bone tissue 1 month after implantation, showing the start of the bioresorption process. On the other hand, Mg content was not found in the specimens 3 and 6 months after implantation, which proves the described hypothesis. The theoretical Ca/P ratio of the HAp (Ca_10_(PO_4_)_6_(OH)_2_) is 1.67. In our study, we achieved a Ca/P of 0.4 at 1 month post-implantation and a Ca/P of >1.5 if the post-implantation time was more than 3 months. Thus, we can indirectly assume that the bioresorption process had completed and the HAp with the indicated composition had instead formed.

### 3.7. Histological Analysis

After 1 month, the “bed” of the implant was clearly visible. In one of the areas, a bone sequester was determined, and it was filled with a dense leukocyte infiltrate with the remains of a homogeneous eosinophilic substance, and perifocal bone resorption was determined with severe fibrosis in the inter-beam space. In other areas, the bone tissue in the area of the bed had not changed (Figure 15a–c).

After 3 months, the “bed” of the implant could be traced throughout and filled with an internal “bone callus” consisting of an outer layer of fibrous tissue with an abundance of osteogenic cells and pronounced angiomatosis of the middle layer of the cartilage tissue (black arrow) and the inner layer of the emerging bone trabeculae (red arrow). In the bone tissue in the area of the bed with foci of incomplete osteogenesis in the form of osteoid formation, we found the presence of proliferating osteoblasts and increased cellularity in the inter-beam space. There was no inflammatory infiltration (Figure 16a–c).

After 6 months, the “bed” of the implant was filled with fibrous tissue, with an abundance of capillaries and a ring-shaped area of compact mature bone. There was no inflammatory infiltration (Figure 17a,b).

## 4. Discussion

Plates, screws, nails, and meshes made of steel or titanium alloys are the gold standard for fracture treatment [44]. According to published statistics, the global market for fracture fixation devices is estimated at USD 5.5 billion [43]. In Russia, on average, 400,000 operations are performed per year using metal structures [45]. However, the use of titanium materials in the treatment of fractures is associated with many disadvantages such as thermal sensitivity [46], tactile sensations of the plates and screws [47,48], limitations in bone growth [47,49], rigidity causing shielding from the stress of the underlying bone [47,50], and the need for repeated surgical interventions in order to remove structures that have fulfilled their roles, which is the primary limitation. In 2018, 176,257 implant removal operations were performed in Germany [51]. This means that metal materials were removed in approximately 80% of the fractures treated with osteosynthesis [52]. The United States reports similar figures [53]. In Germany in 2007, it was estimated that the costs of these procedures exceeded EUR 430 million per year [47,54], while in Russia, this figure amounted to approximately RUB 6 billion [54]. Minimizing the number of such operations will correspondingly reduce the incidence of patients and the financial burden on global health. Our experimental alloy satisfies all the primary mechanical criteria required for bone fixation (tensile yield strength of >230 MPa and tensile strength of >300 MPa) [55]. All the elements (Mg, Zn, and Ga) play important roles in bone metabolism and, in appropriate amounts, can significantly accelerate the healing process [56,57,58].

The composition of the subjects′ blood was analyzed to assess the level of released Mg ions and the markers of inflammation. We did not find a significant increase in serum magnesium levels compared with normal values. Magnesium is regulated in the kidneys, which reabsorb Mg and excrete the excess Mg in the urine [59]. Urinalysis showed that the tested levels of magnesium and creatinine were within normal limits in all animals. These observations are consistent with other studies, which also did not observe increases in the levels of magnesium in the blood sera and did not cause kidney disease [60]. Thus, the results of the analysis of blood and urine allow us to conclude that the degradation of this magnesium alloy did not cause acute, subacute, or chronic systemic inflammatory reactions or pathological changes in the internal organs. This indicates good systemic biocompatibility in vivo. The corrosion of 1 g of pure magnesium produces 1 L of gaseous hydrogen. Several studies have described the presence of gas that was generated as a result of implant degradation [61]. In an in vivo study by Li et al., gas shadows were observed in the soft tissue and bone marrow cavities surrounding a MgCa0.8 implant early after implantation. The gas disappeared two months after implantation, and no side effects were found [62]. This observation is consistent with other studies [63]. Zhang et al. [64] showed that subcutaneous gas bubbles generated by a Mg-6Zn alloy disappeared 6 weeks after implantation, while Hanzi et al. [65] reported limited gas production in the area of a WE43 implant after 91 days. In our study, we investigated possible gas cavities through clinical observation and ultrasound examination. There were no clinically significant gas cavities that could be detected by observation and palpation, and according to the ultrasound results, a gas layer was visualized over the area of installed implants after 3 and 6 months, but after 6 months, it was a significantly smaller volume compared with the same sample after 3 months.

The biodegradation of the implants in vivo was assessed using micro-CT. All implants studied were in direct contact with the surrounding bone and did not show any signs of any adverse reactions. The micro-CT results showed a slight decrease in implant density compared with a non-corroded implant 1 month after implantation. Micro-CT did not reveal the shape of the implant 3 months after implantation. In particular, this study showed that nearly 50% of the implant had degraded after 3 months, and complete bioresorption occurred after 6 months, which was also confirmed by the SEM and EDX analyses. The bone formation around the implants was a very good sign of osseointegration. We can conclude that the post-implantation histological bone formation assessed by micro-CT demonstrated osseointegration and, hence, good biocompatibility with the surrounding bone tissue.

The study of the implant–bone interface using SEM-EDX allowed us to study the interface on a microscopic scale and in terms of chemical composition. Regardless of the implantation period, a thin and compact phosphate-based oxide layer (3–5 µm) was found on the surface of the Mg-based implants. Calcium and magnesium were found in this layer, suggesting the formation of a complex or a mixture of Ca/Mg phosphates and hydroxyapatite under in vivo conditions. Similar results were described in our study, where the implant was in close contact with both the cancellous bone and the cortical layer. In the dynamics of our experiment, and especially after six months, the formation of young bone tissue in the osteotomy zone was visualized, which indicates the complete biodegradation of the installed pins. EDX analysis showed primarily C, Ca, P, and O at 3 and 6 months. We can assume that, 6 months after implantation, the remaining implant consisted primarily of substances similar to the formation of hydroxyapatite. We hypothesize that the carbon content was indicative of the standard mineral constituents of bone, hydroxyapatite Ca_5_(PO_4_)_3_(OH), and calcium phosphate Ca_3_(PO_4_)_2_.

The interaction of the implant with the tissue was studied histologically, and no gas bubbles were observed. This negligible generation of hydrogen gas is a significant advantage over other Mg-based alloys. Hydrogen production is often accompanied by inflammatory reactions and the formation of cavities encapsulated by fibrous tissue [33]. New bone formation has been described for MgCa0.8, AX30, LANd 442, ZEK100, WE43, and LAE442 alloys. Reifenrath et al. [66] also observed a periosteal increase in the rate of mineral intake, which was calculated after in vivo fluorescent labeling [67,68]. Castellani et al. [60] observed greater bone contact with a WE43 magnesium alloy implant compared with a titanium implant. Witte et al. [5] studied four different magnesium alloys and reported a higher mineral attachment rate compared with a degradable implant. These results are correlated with our results. We observed a moderate fit of the bone to the implant. In samples after 6 months, implant beds were observed to be filled with fibrous tissues with an abundance of capillaries and ring-shaped areas of compact mature bone. There was no inflammatory infiltration. The presence of osteoblasts also indicated the ongoing process of bone remodeling. These observations support the idea that Mg-2Zn-2Ga alloy implants are osteoconductive, and they suggest good biocompatibility.

## 5. Conclusions

This study showed that the installation of an implant made of a magnesium alloy (Mg-2Zn-2Ga) did not lead to significant changes in blood parameters and did not lead to the formation of significant gas cavities, and, based on the studies carried out, it can be concluded that there was good biocompatibility and the osteoconductivity of the Mg-2Zn-2Ga was without acute, subacute, or chronic toxic effects. It should be noted that among the radiation methods used for examination, the most informative for assessing the level of resorption was the use of microfocus computed tomography. The next stage planned is to conduct an experimental study on larger animals.

This research will allow various branches of medicine to create the most effective type of fixing structures made of bioresorbable materials consisting of bioneutral and low-toxic elements, which will make it possible to avoid repeated surgical interventions in the future.

## Figures and Tables

**Figure 1 bioengineering-10-00273-f001:**
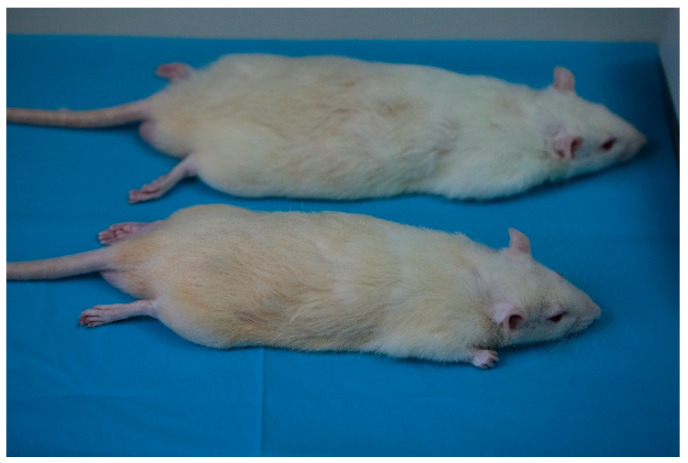
White laboratory rats of the Wistar line.

**Figure 2 bioengineering-10-00273-f002:**
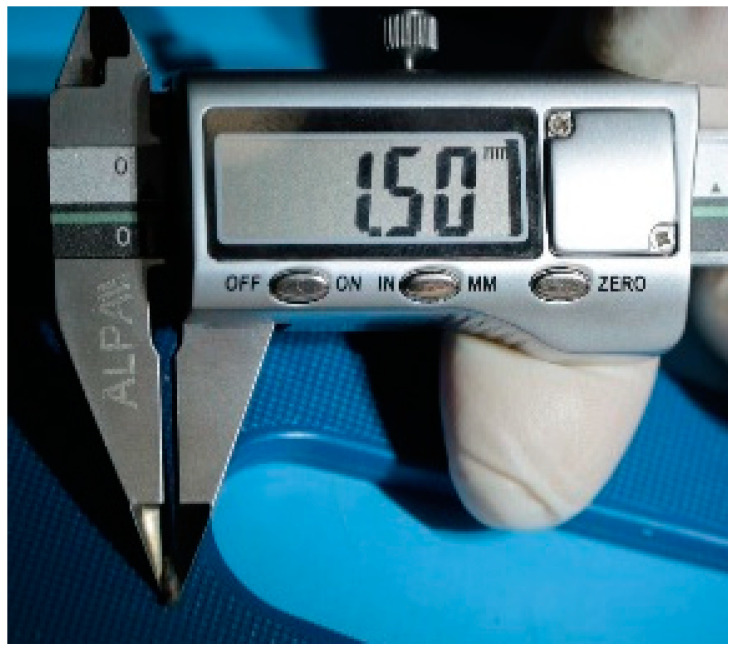
The appearance of the Mg-2Zn-2Ga alloy implants.

**Figure 3 bioengineering-10-00273-f003:**
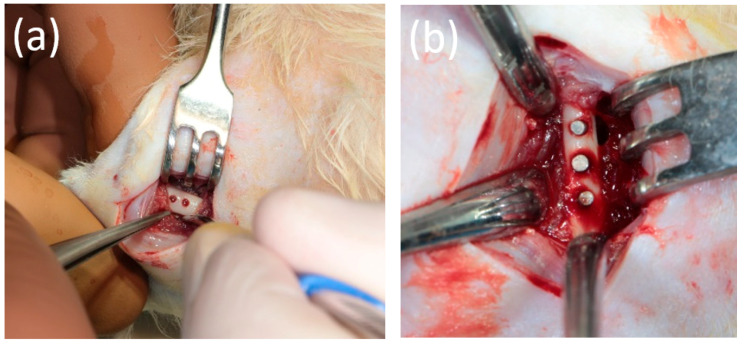
(**a**) The femur of an animal with holes for the implants. (**b**) The appearance of the operating areas after implantation.

**Figure 4 bioengineering-10-00273-f004:**
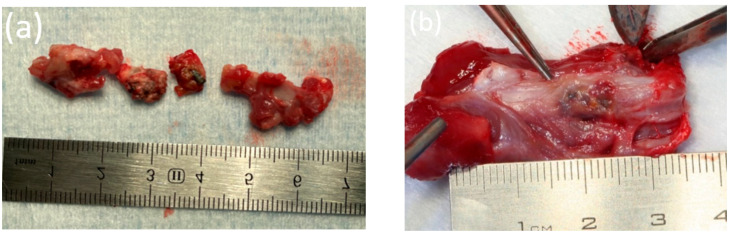
(**a**) Fragments of the femur of an animal 1 month after implantation. (**b**) The femur of an animal 3 months after implantation.

**Figure 5 bioengineering-10-00273-f005:**
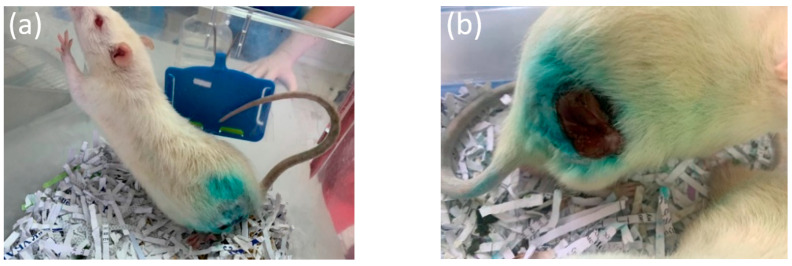
(**a**,**b**) Appearance of the animals and postoperative areas on the third day after implantation.

**Figure 6 bioengineering-10-00273-f006:**
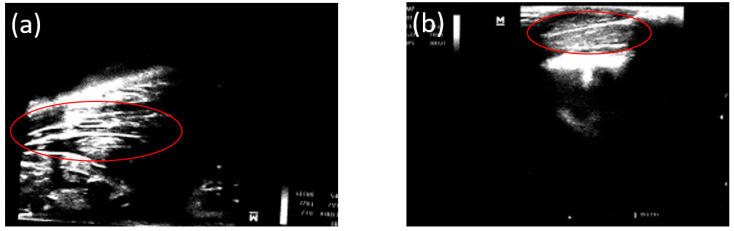
Ultrasound data of the postoperative area after (**a**) 3 months and (**b**) 6 months. The gas layer in the soft tissues was visualized.

**Figure 7 bioengineering-10-00273-f007:**
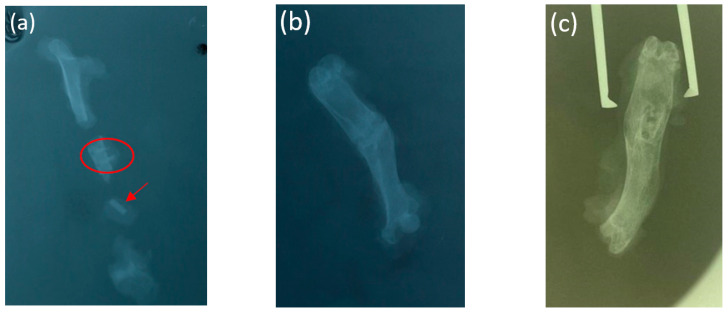
A snapshot of the fragments of the femur of an animal (**a**) 1 month, (**b**) 3 months, and (**c**) 6 months after implantation.

**Figure 8 bioengineering-10-00273-f008:**
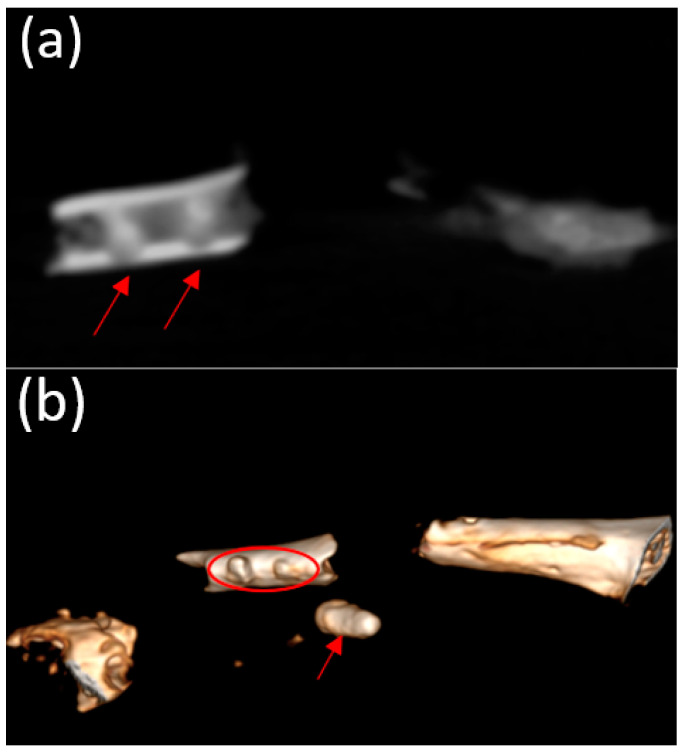
(**a**) Projection of the fragments of the femur of an animal after 1 month. (**b**) CBCT 3D-reconstructed data after implantation.

**Figure 9 bioengineering-10-00273-f009:**
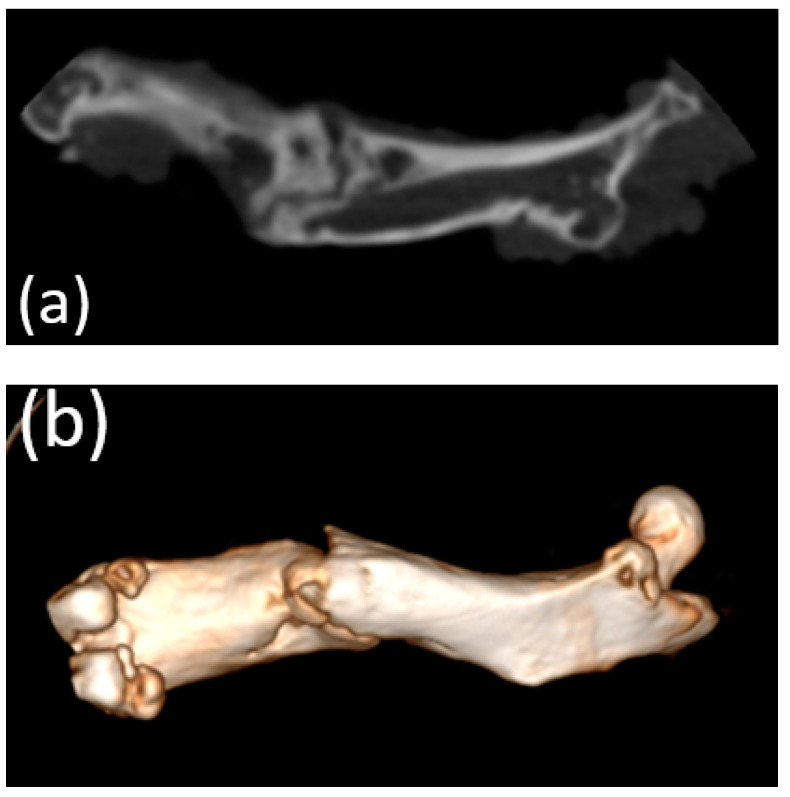
(**a**,**b**) Image of the femur of an animal 3 months after implantation (obtained with CBCT).

**Figure 10 bioengineering-10-00273-f010:**
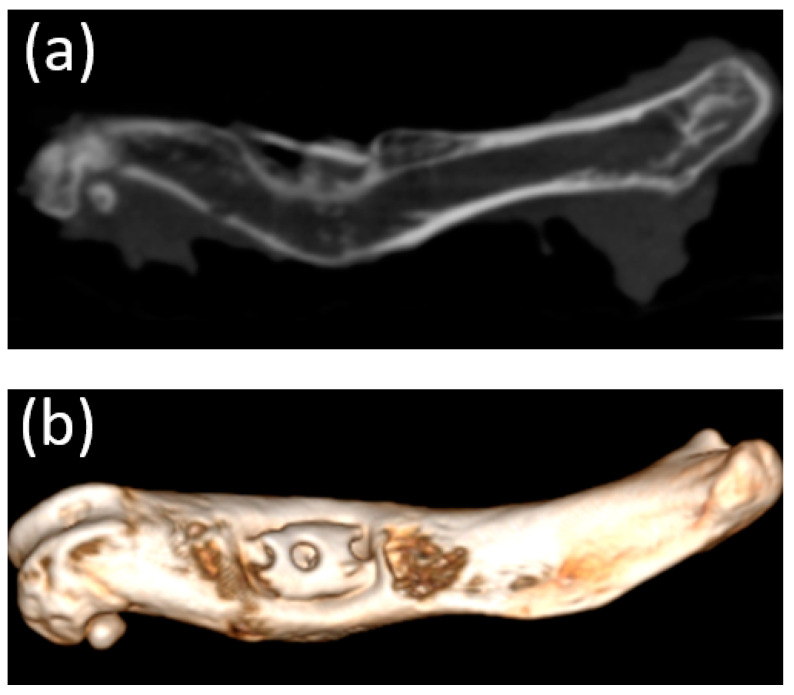
(**a**,**b**) Image of the femur of an animal 6 months after implantation (obtained with CBCT).

**Figure 11 bioengineering-10-00273-f011:**
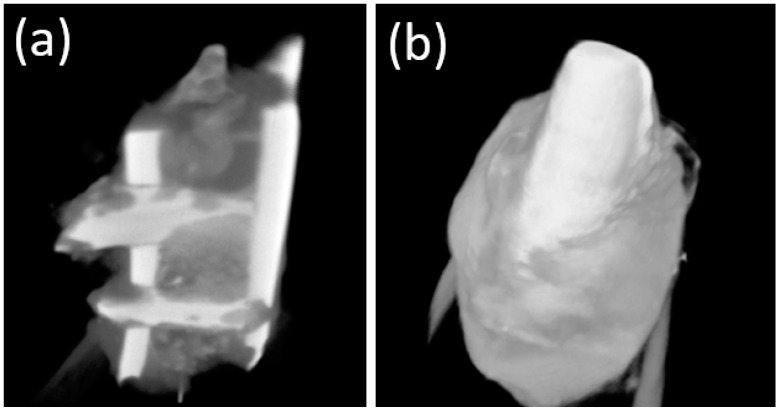
(**a**,**b**) Image of a fragment of the femur of an animal and the implant 1 month after implantation (obtained using micro-CT).

**Figure 12 bioengineering-10-00273-f012:**
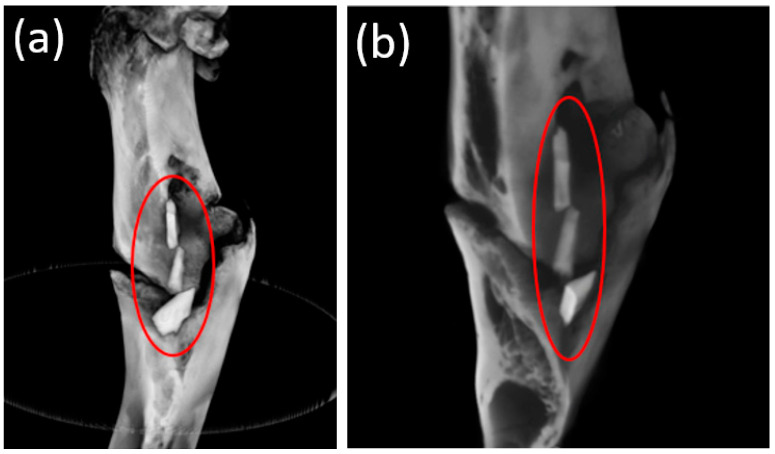
(**a**,**b**) Image of the femur of an animal 3 months after implantation (obtained using micro-CT). A decrease in the average density of the implants was visualized.

**Figure 13 bioengineering-10-00273-f013:**
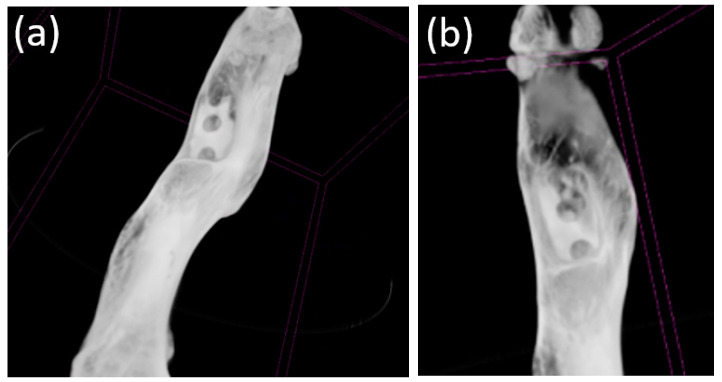
(**a**,**b**) Image of the femur of an animal 6 months after implantation (obtained using micro-CT). Only the holes for the implant placement were visualized; presumably, the implants had been completely resorbed.

**Figure 14 bioengineering-10-00273-f014:**
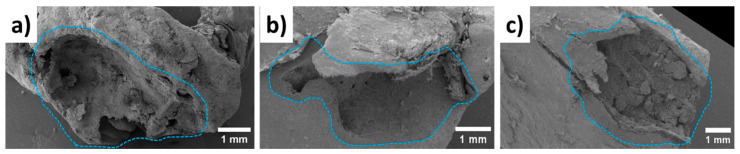
Bones’ microstructures (**a**) 1 month, (**b**) 3 months, and (**c**) 6 months after the operations. The light blue dashed areas indicate the locations of the implants.

**Figure 15 bioengineering-10-00273-f015:**
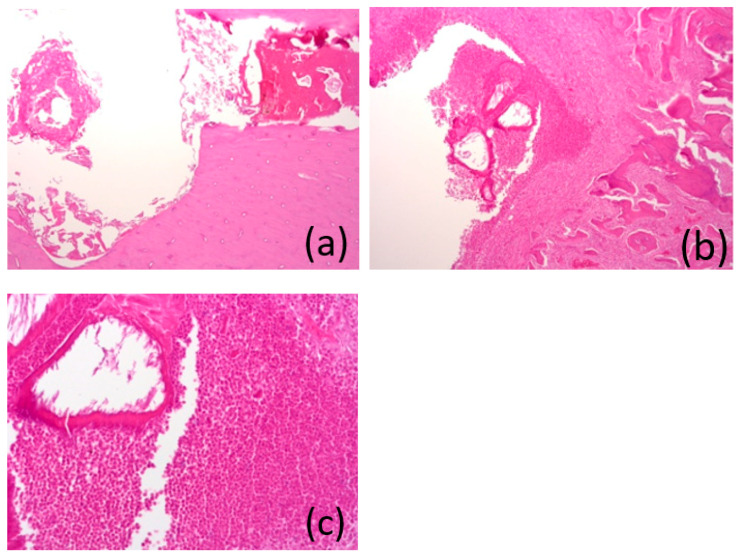
(**a**–**c**) Images of the histological preparations of a sample removed after 1 month and stained with hematoxylin and eosin, using a microscope at magnifications of ×50 (**a**), ×50 (**b**), and ×200 (**c**).

**Figure 16 bioengineering-10-00273-f016:**
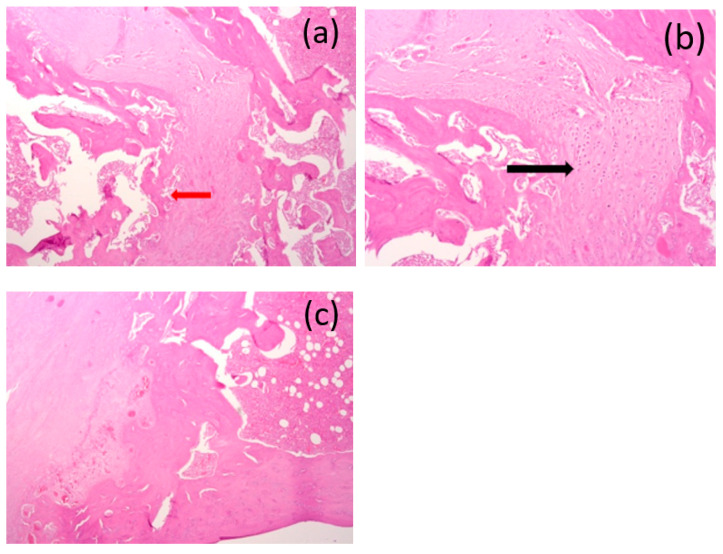
(**a**–**c**) Images of the histological preparations of a sample removed after 3 months and stained with hematoxylin and eosin, using a microscope at magnifications of ×50 (**a**), ×100 (**b**), and ×100 (**c**).

**Figure 17 bioengineering-10-00273-f017:**
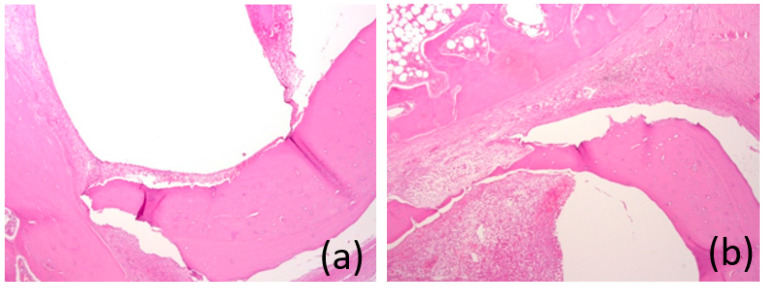
(**a**,**b**) Images of the histological preparations of a sample removed after 6 months and stained with hematoxylin and eosin, using a microscope at ×50 magnification.

**Table 1 bioengineering-10-00273-t001:** Assessment of the levels of resorption of the implants.

	Diameter, mm	Length, mm	Resorption, %	Resorption Rate, mm/week
Before installation	1.70 × 1.70 × 1.70	5.0	-	-
1 month post-operation	1.46 × 1.40 × 1.36	4.6 × 4 × 4	17%	0.04
3 months post-operation	0.7 × 1.0 × 0.8	3.9 × 3.7 × 3.4	49%	0.09
6 months post-operation	Not rendered	Not rendered	100%	0.06

**Table 2 bioengineering-10-00273-t002:** The chemical compositions of the bones near the implant sites.

	Element Concentration, wt.%		
Element Name	After 1 Month	After 3 Months	After 6 Months
Ca	5.32 ± 0.01	3.93 ± 0.01	0.35 ± 0.01
P	13.46 ± 0.02	1.95 ± 0.01	0.23 ± 0.01
Mg	11.80 ± 0.02	-	-
Ca/P	~0.4	~2	~1.5

## Data Availability

Not applicable.

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
