# Peer review of "An In Vivo Rat Study of Bioresorbable Mg-2Zn-2Ga Alloy Implants"

_bioengineering, 2023, doi:10.3390/bioengineering10020273_

Round 1
Reviewer 1 Report
The main problem of current manuscript is unclear novelty. In Introduction It is written that: “The purpose of the present study is to assess the effectiveness of a new Mg alloy 114 bioresorbable implantation system”. The reason for preparation of Mg-2 wt.%Zn-2 wt.%Ga alloy should be explained. Why this composition is chosen? What is the advantages of this alloy over other Mg alloys. The novelty of work have to be cleared in details.
moreover, authors did not use enough references to precisely describe the results and clarify the advantagioes of this work over other studies.
Author Response
Dear Editor and Reviewers!
We are grateful for the Reviewers’ valuable comments concerning the manuscript.
The corrected remarks are in the attached file.
Thank you for your time.
Best regards,
Nikolay Redko

Reviewer 2 Report
This work “Analysis of biodegradable products based on magnesium alloys for medical usage: experimental study on rats” is devoted to the magnesium alloy implant. This material has good biocompatibility. Microfocus computed tomography was used for control. This will practically make it possible to create an effective type of fixing structures from bioneutral and low-toxic materials for various fields of medicine.
• The work is written clearly and distinctly.
• Graphs and formulas are clear and informative.
• The work can be published in the presented version.
Conclusion:
The work can be published in the presented version.
Author Response

(The authors gave the same response as above.)

Reviewer 3 Report
Line no 21-24 is not relevant. The abstract must be written only with experimental results. Kindly modify it. May consider to include the salient findings .
The introduction is perfectly fine.
How come the oxide layer prevention take care while melting the alloys. Do the author have any SEM EDX or XRD analysis available to value added the work?
The solution heat treated conditions are not available detailed. Kindly include. The weight of the animal can be mentioned in the text. Fig 2 must be changed. Kindly include the scale before the pin and take the images.
Figure 4a) and b is not marked in the image.
Check line no 194 which starts with t. Similarly line no 212. 2.7 heading can be renamed. historical evaluation of what?
Mark Fig 5a) to c) in the manuscript figure. There are plenty of figure and the authors observed and reported the results and not discussed scientifically. Check line no 400. Why the authors jump in to aim again.
Line no 446, we can assume.
I request the 1st and corresponding author to sit together , edit the paper with more no of discussion and re submit the article. The language must edit from the native english speakers.
Author Response

(The authors gave the same response as above.)

Round 2
Reviewer 3 Report
Dear authors,
In the results section around line no 255 remove the red colur highlighted a) . Kindly allign it properly.Similarly for figure 11-13.
Author Response
Dear Reviewer!
Thank you for your comments and edits. All technical moments have been eliminated!
Best regards,
Nikolay Redko